# Sparse-view Pose Estimation and Reconstruction via Analysis by Generative Synthesis

**Qitao Zhao**    **Shubham Tulsiani**
Carnegie Mellon University
*Project page:* qitaozhao.github.io/SparseAGS

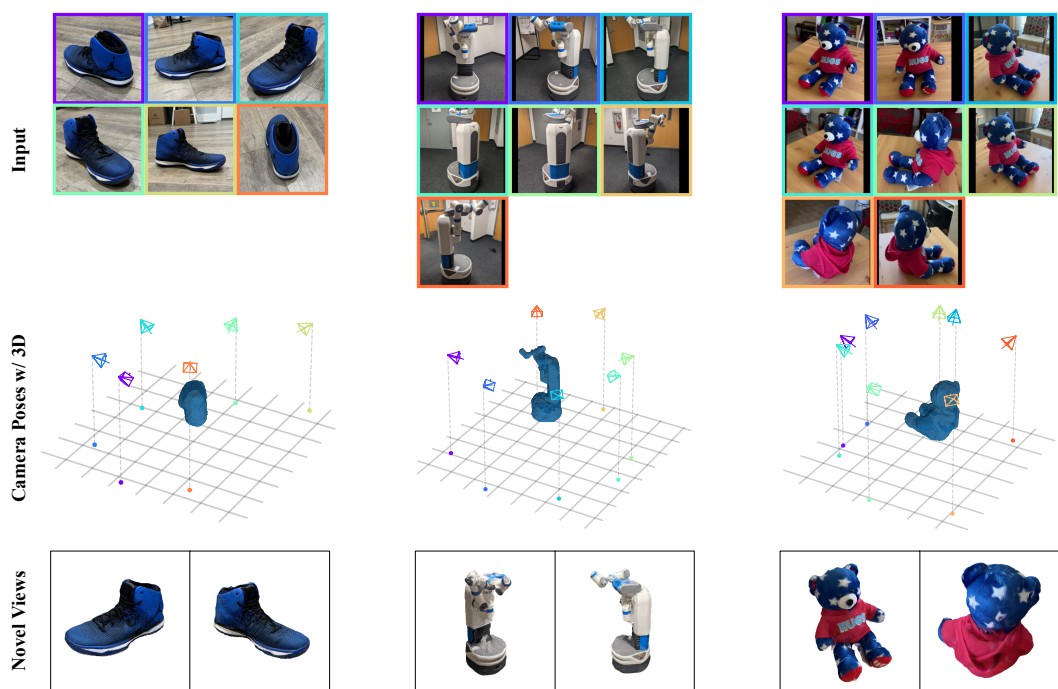

Figure 1: Given a set of unposed input images, SparseAGS jointly infers the corresponding camera poses and underlying 3D, allowing high-fidelity 3D inference in the wild.

## Abstract

Inferring the 3D structure underlying a set of multi-view images typically requires solving two co-dependent tasks – accurate 3D reconstruction requires precise camera poses, and predicting camera poses relies on (implicitly or explicitly) modeling the underlying 3D. The classical framework of analysis by synthesis casts this inference as a joint optimization seeking to explain the observed pixels, and recent instantiations learn expressive 3D representations (*e.g.*, Neural Fields) with gradient-descent-based pose refinement of initial pose estimates. However, given a sparse set of observed views, the observations may not provide sufficient direct evidence to obtain complete and accurate 3D. Moreover, large errors in pose estimation may not be easily corrected and can further degrade the inferred 3D. To allow robust 3D reconstruction and pose estimation in this challenging setup, we propose SparseAGS, a method that adapts this analysis-by-synthesis approach by: a) including novel-view-synthesis-based generative priors in conjunction with photometric objectives to improve the quality of the inferred 3D, and b) explicitly

38th Conference on Neural Information Processing Systems (NeurIPS 2024).

reasoning about outliers and using a discrete search with a continuous optimization-based strategy to correct them. We validate our framework across real-world and synthetic datasets in combination with several off-the-shelf pose estimation systems as initialization. We find that it significantly improves the base systems' pose accuracy while yielding high-quality 3D reconstructions that outperform the results from current multi-view reconstruction baselines.

# 1   Introduction

Consider the images of the robot shown in Fig. 1. From just these few images, we humans can easily understand the 3D structure of this object – it has a cylindrical base supporting a tall body from which an arm extends to the front. We do this by aggregating the information across images into a consistent 3D mental model, *e.g.*, the "front" view informs us of the width of the body and the "side" view(s) about the extended arm. But how do we know which image is to the "front" or to the "side" to begin with? As evidenced in seminal research of mental rotation [30], we understand viewpoints by forming mental 3D models. Thus, to form mental 3D models, we need to understand the (relative) viewpoints across images, but doing so in turn requires a mental 3D model!

This co-dependency in inferring shape and pose is one that any computational approach aiming to recover 3D from multiple views also needs to deal with. Indeed, classical approaches like Structure-from-Motion (SfM) [29] tackle the two together and infer 3D and camera viewpoints. However, these correspondence-based methods can only infer sparse 3D representations and are not robust given a small set of images with limited overlap. To allow 3D inference in such sparse-view settings, recent learning-based approaches have pursued sparse-view reconstruction approaches [52, 45], but assuming known precise camera poses. Separately, there have been several methods [18, 47, 48, 31] which predict camera viewpoints given a set of images. While these methods have led to impressive results for both 3D reconstruction and pose inference, their singular focus on only one task without tackling the other limits their utility – the 3D reconstruction methods requiring precise cameras cannot be easily used in real-world applications, and pose estimation methods that do not model 3D are typically limited in their accuracy.

We present SparseAGS, a framework that unifies the advances in learning-based pose estimation and 3D reconstruction – using the former as an initialization and building on the latter for obtaining accurate 3D reconstruction. Specifically, we adopt an "analysis by synthesis" approach where we jointly optimize pose and 3D to explain the observed pixels. However, unlike prior methods [19, 42] which simply leverage photometric-error-driven gradient-based updates for pose and 3D, we additionally leverage generative priors [8, 33] for more complete (and accurate) 3D despite input images that may only partially capture the object. However, current off-the-shelf novel-view generative models [20] only allow 3-DoF camera parameterization which is insufficient beyond synthetic settings, we finetune a SoTA model to allow 6-DoF camera variation when querying novel views. We find that such generative priors not only contribute to the 3D reconstruction quality but also result in more accurate camera poses. Moreover, we also explicitly account for large possible errors in initial camera estimation and prevent these from degrading 3D reconstruction via identifying outliers, and also improve poses via a combination of a discrete search and continuous optimization.

Compared to prior joint reconstruction and pose estimation methods that are designed to improve near-perfect initial cameras [19, 38], SparseAGS can leverage off-the-shelf pose estimates, thereby allowing robust inference in real-world scenarios. We demonstrate its efficacy using both, real-world and synthetic datasets in conjunction with several state-of-the-art pose estimation methods as initialization. We find that our approach significantly improves the initial camera estimates and yields high-fidelity 3D reconstructions (and novel view renderings). In summary, our contributions are:

- We introduce an analysis-by-generative-synthesis framework that jointly estimates 3D and camera viewpoints given a sparse set of input images, by integrating a 6-DoF novel-view generative prior in an analysis-by-synthesis approach
- Our approach allows leveraging any off-the-shelf pose estimation system and can robustly estimate 3D and viewpoints despite large errors in the initial estimates.
- We present results across datasets and initializations and show clear improvements over the initializations as well as outperform prior sparse-view 3D reconstruction baselines.

## 2 Related Work

**Sparse-view Pose Estimation**. Traditional correspondence-based Structure-from-Motion [32, 29] methods often fail to estimate camera poses in sparse-view settings. Several approaches instead seek to leverage data-driven priors, for example learning energy-based [48, 18] or denoising diffusion [39] models to predict cameras. While these approaches predict global camera representations, some works have demonstrated the benefit of denser camera parametrizations by predicting raymaps [47] or pointmaps [41, 17]. As an alternative paradigm to direct camera prediction, some recent methods [3, 43, 34] instead estimate relative poses by inverting the view-conditioned synthesis capabilities of diffusion models [20]. While these methods have led to remarkable improvements in camera estimation, these are still susceptible to some imprecision and occasional outliers which our 3D-reasoning-based approach can correct.

**Sparse-view 3D Reconstruction**. This line of work aims to recover 3D from sparsely sampled views, aiming to infer complete 3D representations that faithfully reflect the content captured by the input images while making reasonable guesses for invisible areas. The progress of diffusion models [8, 33] has greatly advanced this direction, as they are capable of learning strong natural image priors from data. Inspired by DreamFusion [23], which generates 3D scenes given textual descriptions leveraging a text-to-image diffusion model [26], SparseFusion [52] learns a view-conditioned diffusion model on multi-view image collections for novel view synthesis and then distills the learned novel-view distributions into a single consistent 3D representation. DreamSparse [45] further improves the performance by utilizing internet-scale natural image priors learned by Stable Diffusion [25]. Although these methods present impressive results, they assume precise camera poses are available, which limits their applications. FORGE [11] addresses this by jointly inferring both camera poses and 3D structure in a single forward pass, though the quality of its reconstructions remains constrained by pose estimation accuracy without further refinement or correction.

**Pose-free Sparse-view 3D Reconstruction**. Some recent works [27, 40, 12, 13] have attempted to bypass the reasoning about camera poses and directly infer novel views or 3D representations from unposed images. An unposed variant of the Scene Representation Transformer [27] encodes a set of input images into latent features and synthesizes novel views given the corresponding query rays (w.r.t. the viewpoint of the first image) using a transformer encoder and decoder. UpFusion [13] improved upon this by learning a diffusion model and distilling a consistent 3D representation via Score Distillation Sampling [35], whereas LEAP [12] and PF-LRM [40] can directly predict (volumetric or triplane) 3D representations in a feedforward manner. While these methods demonstrate promising results, their geometry-free approach cannot easily capture the specific details across input images and they struggle to improve the 3D estimation with additional input images.

**Analysis-by-synthesis Approaches**. Approaching visual perception as an inverse graphics task is classical idea in computer vision [15, 46], and has been leveraged for inferring scene properties (*e.g.*, object pose) by synthesizing visual content as close to observations as possible [2, 16, 51, 50]. Closer to our setup, prior approaches jointly optimize camera pose and 3D representation (*e.g.*, NeRF [22]) to explain the observed images [19, 42] but these methods are designed for dense observations and only handle small pose errors. Closer to our work, SPARF [38] focuses on the sparse-view setup, leveraging estimated pixel correspondence [37] as prior knowledge in addition to the standard photometric loss. However, reliably extracting such correspondences can be challenging, and false match estimates may even confuse pose refinement, leading to degraded 3D reconstruction.

## 3 Method

### 3.1 Overview

**Analysis by Synthesis**. Given a set of sparse-view images, denoted as $\mathbb{I} \equiv \{\boldsymbol{I}_i\}_{i=1}^N$, our goal is to reconstruct the underlying 3D structure $\theta$ and infer the camera poses corresponding to the input images $\Pi \equiv \{\boldsymbol{\pi}_i\}_{i=1}^N$. This can be done by solving an *analysis-by-synthesis* problem

$$\min_{\theta, \Pi} \sum\nolimits_{i=1}^N ||\boldsymbol{I}_i - f_\theta(\boldsymbol{\pi}_i)||^2 \tag{1}$$

where $f_\theta$ is a rendering function. Eq. 1 demonstrates that we want to find a scene description consisting of a 3D representation $\theta$ and camera configurations $\Pi$ that well explain the observed input

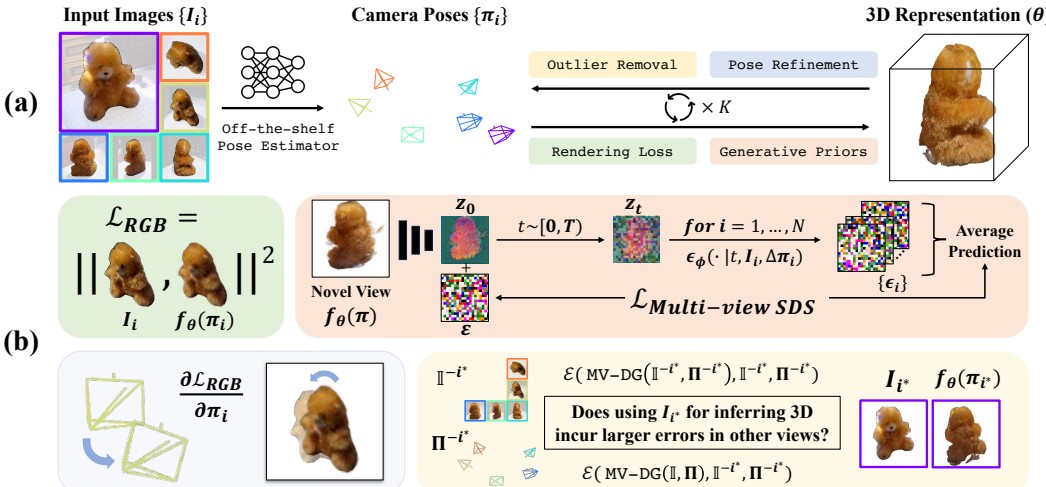

Figure 2: **(a) Overview of SparseAGS:** Given estimated camera poses from off-the-shelf models, our method iteratively reconstructs 3D and optimizes poses leveraging diffusion priors. **(b) Detailed View of Each Component:** We use rendering loss and multi-view SDS loss for 3D reconstruction while the rendering loss is propagated back to refine camera poses. At the end of each reconstruction iteration, we identify outliers by checking if their involvement in 3D inference yields larger errors in other views, implying the inconsistency of their poses with others.

images. If $f_\theta$ is differentiable, we can jointly optimize the 3D representation and camera poses via gradient descent [19, 42]. However, this approach may not work well in the sparse-view setting [38] (*i.e.*, $N$ is small) as the 3D representation can overfit to the input images without forming a plausible structure, degrading both, pose estimation and 3D reconstruction.

**Analysis by Generative Synthesis**. To address this issue, we propose to introduce generative priors into analysis by synthesis, so we term our method *analysis by generative synthesis*. In addition to the known-view objective (Eq. 1), we leverage diffusion priors [8, 33] to optimize renderings from randomly sampled novel views ($\pi$) as well

$$\min_\theta \mathbb{E}_\pi \; -\log p_\phi(f_\theta(\pi)|\pi, \mathbb{I}, \Pi) \tag{2}$$

where $p_\phi$ is the likelihood of the novel view rendering conditioned on the viewpoint $\pi$ and inputs $(\mathbb{I}, \Pi)$, modeled by the diffusion model $\phi$. The gradients for this objective can be obtained via Score Distillation Sampling (SDS) [23], and intuitively, they encourage the renderings of the 3D representation to be plausible based on image distributions learned by the diffusion model.

In the following, we first introduce a few preliminaries about an efficient single-view-to-3D approach (Sec. 3.2), on which we build our multi-view reconstruction method, MV-DreamGaussian, enabling analysis by generative synthesis in the wild (Sec. 3.3). Then, we present our complete framework that involves dealing with imprecise cameras (Sec. 3.4). An illustration of our approach is in Fig. 2.

### 3.2 Preliminaries: DreamGaussian

DreamGaussian [36] generates 3D from a single image with a two-stage approach, achieving a satisfactory trade-off between speed and fidelity. The first stage optimizes 3D Gaussians [14] (parameterized by $\theta$) using a combination of photometric loss (Eq. 1, except that the camera pose is not optimized) and SDS loss (Eq. 3) with a view-conditioned diffusion model, Zero-1-to-3 [20]. Specifically, for a randomly sampled novel view $\pi$, scheduled noise $\epsilon$ at timestep $t$ is added to the latent of its rendering (the noisy latent is denoted by $\mathbf{z}_t$). The training objective minimizes the difference between the predicted noise and the added noise, approximating the negative log-likelihood of the rendered image. The gradient of SDS loss is given by

$$\nabla_\theta \mathcal{L}_{\text{SDS}} = \lambda_{\text{SDS}} \mathbb{E}_{t,\pi,\epsilon} \left[ w(t)(\epsilon_\phi(\mathbf{z}_t; t, \boldsymbol{I}_1, \Delta\pi) - \epsilon) \frac{\partial f_\theta(\pi)}{\partial \theta} \right] \tag{3}$$

where $w(t)$ is a weighting function, $\epsilon_\phi(\cdot)$ is a U-Net trained to predict the added noise given the noisy latent $\mathbf{z}_t$, conditioned on the timestep $t$, reference image $\boldsymbol{I}_1$, and the relative camera pose $\Delta\boldsymbol{\pi}$. This stage efficiently builds the geometry of the object with rough texture, which takes 500 training steps (in about 1 minute). In the second stage, 3D Gaussians are converted to a textured mesh with Marching Cubes [21], and only its texture is optimized. This stage takes another 50 steps and can finish within 30 seconds on a single GPU.

We find DreamGaussian to be a suitable starting point to perform *analysis by generative synthesis*, but note that it has some key limitations: (1) **3-DoF Camera Parameterization**. DreamGaussian adopts a 3-DoF camera parameterization (*i.e.*, radius, elevation, and azimuth) to accommodate the camera definition in Zero-1-to-3 [20]. While this parameterization is sufficient for synthetic data, it cannot well represent the 6-DoF camera poses of real-world images. (2) **Single Input Image**. DreamGaussian is designed for the singe-view-to-3D task. In contrast, we aim for the reconstructed 3D to reflect the details captured by multiple input images faithfully. This requires an approach to handling information from multi-view images.

### 3.3 Leveraging Generative Priors for Sparse-view 3D in the Wild

We adapt DreamGaussian's two-stage method and extend it to (1) handle real-world images with 6-DoF camera parameters and (2) utilize sparse-view images as input.

**Generative Priors in the Wild**. Zero-1-to-3 [20] offers desirable generative priors that enable single-view-to-3D generation of DreamGaussian. However, it assumes no in-plane camera rotation and that all possible camera poses are strictly directed toward a common origin. We find these assumptions are over-restrictive for real-world images. Therefore, we propose to replace the 3-DoF camera condition in Zero-1-to-3 with a 6-DoF one, represented as an 18-dimensional vector:

$$[\text{Flatten}(\boldsymbol{\pi}_{\text{rel}}), \log(f^x_{\text{rel}}), \log(f^y_{\text{rel}})] \tag{4}$$

where $\boldsymbol{\pi}_{\text{rel}}$ is the relative extrinsic matrix ($4\times4$) between the source view and target view, and $f^x_{\text{rel}}$ ($f^y_{\text{rel}}$) is the ratio of the focal length along the $x$- ($y$-) axis between them. We include the focal length term to account for the object scale change due to cropping. This simple camera parameterization effectively represents 6-DoF cameras in the wild. Details regarding finetuning Zero-1-to-3 for 6-DoF camera conditioning are deferred to Sec. C in the appendix. We note that recent work, ZeroNVS [28], also discussed this 3-DoF issue of Zero-1-to-3 and proposed a "6-DoF+1" camera parameterization for scene-level novel view synthesis. However, this approach is not directly applicable to our object-centric setting, as it is trained using images with complex backgrounds and leverages depth priors to address scale ambiguity.

**Leveraging the Generative Priors from Multiple Views**. DreamGaussian only uses the generative priors from a single reference image via SDS loss. To make $\mathcal{L}_{\text{SDS}}$ aware of the visual cues from multiple input images, we modify Eq. 3 as

$$\nabla_\theta \mathcal{L}_{\text{Multi-view SDS}} = \lambda_{\text{SDS}} \, \mathbb{E}_{t,\boldsymbol{\pi},\boldsymbol{\epsilon}} \left[ w(t)(\overline{\boldsymbol{\epsilon}}_\phi - \boldsymbol{\epsilon}) \frac{\partial f_\theta(\boldsymbol{\pi})}{\partial \theta} \right], \text{where} \tag{5}$$

$$\overline{\boldsymbol{\epsilon}}_\phi = \frac{1}{N} \sum_{i=1}^{N} \boldsymbol{\epsilon}_\phi(\mathbf{z}_t; t, \boldsymbol{I}_i, \Delta\boldsymbol{\pi}_i) \tag{6}$$

$N$ is the total number of input views, $\boldsymbol{I}_i$ is the $i^{th}$ input image, and $\Delta\boldsymbol{\pi}_i$ is its relative camera pose w.r.t. the sampled novel view $\boldsymbol{\pi}$. We average the noise predictions from all input views that share the same timestep $t$. This method draws inspiration from the implementation of Stable-Dreamfusion [35], but we do not weigh the predicted noises based on the relative closeness of camera poses. The rationale behind this is that the camera poses in our setting are not always reliable, and relying too heavily on "close" views could introduce significant conflicts during the 3D optimization process.

With these modifications, our multi-view reconstruction approach, termed *MV-DreamGaussian*, is capable of reconstructing 3D from sparse images in the wild by leveraging diffusion priors. When describing its use in our overall framework, we use the notation $\theta = \text{MV-DG}(\mathbb{I}, \Pi)$ to denote the 3D representation inferred via this pipeline given a set of input images $\mathbb{I}$ and associated viewpoints $\Pi$.

### 3.4 3D Reconstruction with Imperfect Cameras

Here, we introduce the complete framework of SparseAGS (see Fig. 2 for an overview) that: a) leverages off-the-shelf pose estimation methods and b) incorporates our multi-view reconstruction

approach MV-DG (Sec. 3.3) to jointly infer accurate 3D and camera viewpoints. A key challenge we seek to overcome is that the estimated camera viewpoints may have significant errors and that naively using all images to infer 3D can result in suboptimal estimates.

**Pose Refinement via Gradient Descent**. During 3D reconstruction via MV-DreamGaussian, we back-propagate gradients from the photometric loss (Eq. 1) back to update camera poses (implementing custom CUDA kernels to enable this gradient computation). This process allows the camera poses to become more precisely aligned as 3D reconstruction progresses. We denote by $\Pi' = \text{GD}(\mathbb{I}, \theta, \Pi)$ the resulting camera viewpoints from this optimization given the set of input images $\mathbb{I}$, and 3D reconstruction $\theta$ and initial poses $\Pi$.

With this pose-and-3D co-optimization, we can instantiate a version of our analysis-by-generative-synthesis framework by iteratively refining poses and reconstructing 3D given initial pose estimates $\Pi_0$ from an off-the-shelf system:

$$\text{For } k = 1 \cdots K: \quad \theta_k = \text{MV-DG}(\mathbb{I}, \Pi_{k-1}); \quad \Pi_k = \text{GD}(\mathbb{I}, \theta_k, \Pi_{k-1}) \tag{7}$$

For clarity, we present separate formulas for the reconstructed 3D $\theta_k$ and the updated poses $\Pi_k$, though they are part of the same optimization process. Notably, in each iteration, we initialize the camera poses using the output from the previous iteration ($\Pi_{k-1}$), while the 3D representation ($\theta_k$) is reset and reconstructed from scratch.

**Dealing with Outliers**. Although the above iterative optimization framework can allow us to infer consistent poses and 3D reconstructions, it is susceptible to local optima and not robust to large errors in initial pose estimates. To overcome this, we additionally detect *"outliers"*, *i.e.*, images with possibly large pose errors that degrade the quality of 3D reconstruction. We then modify our approach to leverage only the estimated inliers for 3D reconstruction while also separately performing a discrete search to update the outlier viewpoints.

*Iterative Outlier Identification.* Our key insight is that an "outlier" image not only exhibits high reprojection error, making it difficult to reconstruct on its own, but also that including it as a training image for 3D reconstruction degrades the overall quality, thus leading to poorer reconstruction even from other views! We operationalize this insight by classifying an image as an outlier if removing it from training significantly improves performance on other images. More formally, let $\mathbb{I}^{-i}$ denote the set of images after removing the $i^{th}$ one and let $\mathcal{E}(\theta, \mathbb{I}, \Pi)$ denote the average reprojection error of a 3D representation $\theta$ over images $\mathbb{I}$ with (predicted) poses $\Pi$. We consider an image $i$ as an outlier if

$$\mathcal{E}(\text{MV-DG}(\mathbb{I}, \Pi), \mathbb{I}^{-i}, \Pi^{-i}) >> \mathcal{E}(\text{MV-DG}(\mathbb{I}^{-i}, \Pi^{-i}), \mathbb{I}^{-i}, \Pi^{-i}) \tag{8}$$

*i.e.*, adding the image to training set significantly increases the error for other views. For efficiency, instead of considering all images as outlier candidates, we iterate over images in decreasing order of reprojection error. Given this procedure to detect outliers, at each iteration (except $k = 1$) we modify the above framework to first filter out the outliers found in previous iterations (along with the new *"outlier candidate"* that gives the largest reprojection error at the last iteration):

$$\mathbb{I}_{k-1}^{\text{inlier}}, \Pi_{k-1}^{\text{inlier}} \equiv \text{filter-outliers}(\mathbb{I}, \theta_{k-1}, \Pi_{k-1}) \tag{9}$$

and only use the estimated inliers for optimizing 3D: $\theta_k = \text{MV-DG}(\mathbb{I}_{k-1}^{\text{inlier}}, \Pi_{k-1}^{\text{inlier}})$. This filter-and-reconstruct loop stops when either the selected outlier candidate is determined to be an inlier (*i.e.*, the condition 8 is not satisfied) or the number of remaining inliers falls below a threshold (*e.g.*, 4).

*Correcting Outlier Poses.* While identifying the outliers allows us to prevent them from influencing the 3D inference, the finally recovered model may not capture the details from all images. We thus also attempt to "correct" the pose estimates for the outliers via a discrete search (followed by continuous optimization). Using the currently estimated 3D (reconstructed from only the inliers), for each image in the outlier set, we re-estimate its camera pose via render-and-compare. We first densely sample pose candidates on a sphere and render images from the current 3D. We rank the pose candidates by measuring both pixel-space error (*i.e.*, MSE) and perception error (*i.e.*, LPIPS [49]). The pose candidate with the highest cumulative rank is selected as the optimal solution. Once all identified outliers are corrected, another reconstruction is performed to form a consistent 3D representation with all images using the updated poses.

Our overall framework is very efficient (largely due to an efficient implementation of the reconstruction step), typically taking 5-10 minutes given 8 input images, with increased inference time depending on the number of estimated outliers. We include a brief analysis of the inference time of our system in Sec. B.

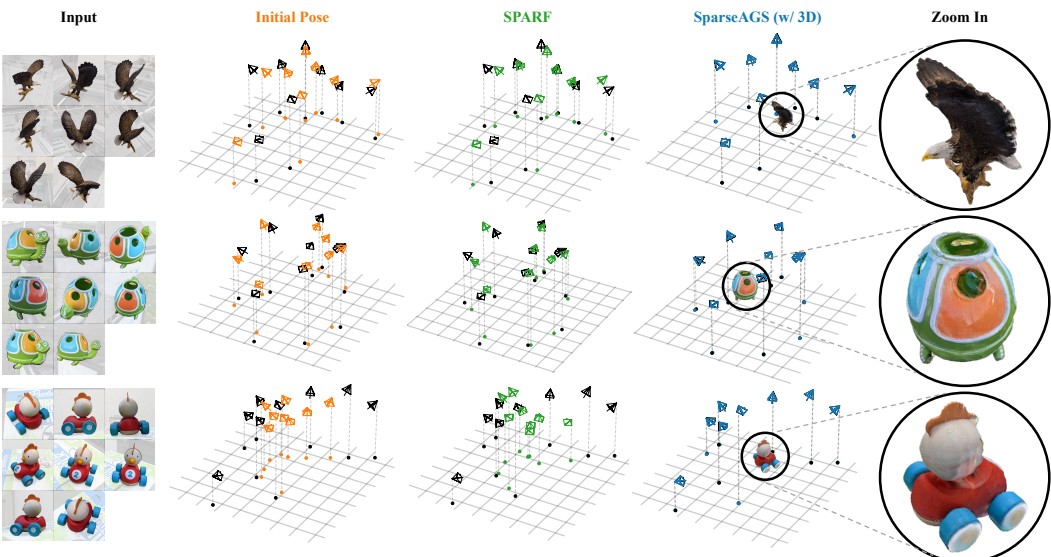

Figure 3: **Qualitative Comparison on Camera Pose Accuracy.** Given initial poses from off-the-shelf methods (top to bottom: DUSt3R [41], Ray Diff. [47] and RelPose++ [18]), the refined poses from SPARF [38] are compared with the output of SparseAGS. The estimated cameras are aligned with ground truth (in black) with an optimal similarity transform. More results are available in Fig. 8.

## 4 Experiments

### 4.1 Experimental Setup

**Datasets**. We primarily evaluate our method on a real-world multi-view object-centric dataset NAVI [9]. This dataset includes high-quality foreground masks, precise camera poses, and 3D meshes. For each of the 35 objects in NAVI, we randomly select 5 multi-view sequences for pose estimation and reconstruction. Additionally, we assess our method on synthetic datasets, including GSO [7], ABO [4], and OmniObject3D [44]. Results for the synthetic datasets are provided in Sec. E of the appendix.

**Baselines**. To evaluate camera pose accuracy, we select three sparse-view pose estimation baseline methods: RelPose++ [18], Ray Diffusion [47], and DUSt3R [41]. The first two are trained exclusively

Table 1: **Comparison of Camera Rotation and Center Accuracy with SPARF [38].** We use three pose estimation baselines (RelPose++ [18], Ray Diff. [47], DUSt3R [41]) and measure rotation accuracy at two thresholds (5 and 15 degrees) and camera center accuracy at a threshold of 0.1 (of the scene scale). Eight images are used.

| Method | Rot.@5° ↑ | Rot.@15° ↑ | CC@0.1 ↑ |
|---|---|---|---|
| **RelPose++** | 10.9 | 56.4 | 26.0 |
| w/ SPARF | 28.6(**+17.7**) | 51.9(**-4.5**) | 37.9(**+11.9**) |
| w/ SparseAGS | 42.1(**+31.2**) | 67.6(**+11.2**) | 53.3(**+27.3**) |
| **Ray Diff.** | 13.5 | 73.5 | 38.3 |
| w/ SPARF | 46.0(**+32.5**) | 76.1(**+2.6**) | 65.8(**+27.5**) |
| w/ SparseAGS | 60.3(**+46.8**) | 88.2(**+14.7**) | 80.4(**+42.1**) |
| **DUSt3R** | 52.3 | 93.8 | 82.2 |
| w/ SPARF | 59.7(**+7.4**) | 87.8(**-6.0**) | 81.9(**-0.3**) |
| w/ SparseAGS | 83.7(**+31.4**) | 96.2(**+2.4**) | 93.5(**+11.3**) |

on CO3D [24], while DUSt3R is trained on a mixture of eight datasets, representing different levels of precision in initial camera poses. Our method initializes and improves the pose estimates from these baselines, and we also compare with SPARF [38], a sparse-view pose-NeRF co-optimization method. For evaluation of novel view synthesis, we mainly compare our method with unposed sparse-view reconstruction approaches, LEAP [12] and UpFusion [13] (we include comparison with SPARF in Sec. D). We conduct experiments with varying numbers of input images (N = 6, 8, 10, 16).

**Metrics**. For pose accuracy, we follow prior works [18, 47] and report the following metrics: (1) Rotation accuracy: we compare pairwise relative rotation between the predicted cameras and ground truth. We report the proportion of samples with errors less than a specified threshold, such as 5 and 15 degrees. (2) Camera center accuracy: we align the predictions and ground truth using an optimal similarity transform and report the proportion of camera centers within 10% of the scene

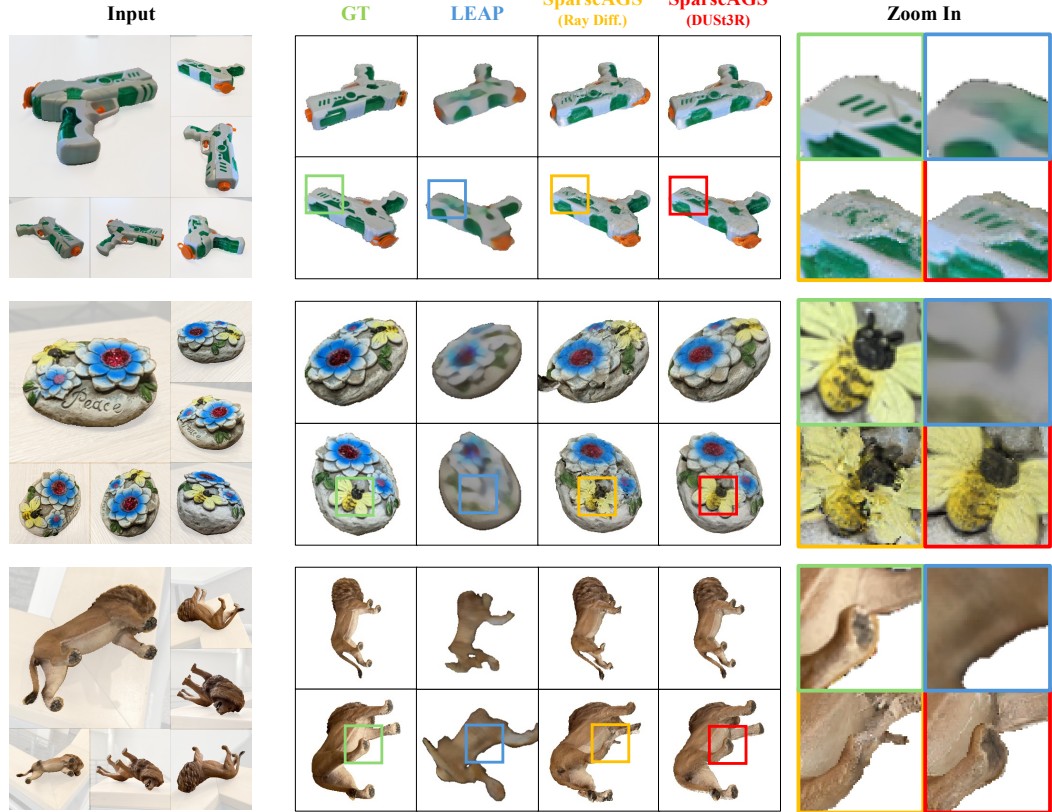

Figure 4: **Qualitative Comparison with LEAP [12] on Novel View Synthesis.** We use two pose estimation baselines (Ray Diffusion [47] and DUSt3R [41]). SparseAGS better preserves details from the input images and shows enhanced performance with more accurate initial camera poses. More results are available in Fig. 9 of the appendix.

scale relative to the ground truth. We evaluate our 3D representation via novel-view synthesis and report PSNR and LPIPS [49] for the rendered views. In our ablation study, we also assess the 3D geometry using the F1 score, comparing our recovered geometry against the ground truth 3D meshes.

Table 2: **Evaluation of Camera Pose Accuracy with Varying Numbers of Input Images on NAVI [9].** Here we use the same evaluation protocols as Tab. 1.

| Method | N = 6 | | | N = 10 | | | N = 16 | | |
|---|---|---|---|---|---|---|---|---|---|
| | Rot.@5° | Rot.@15° | CC@0.1 | Rot.@5° | Rot.@15° | CC@0.1 | Rot.@5° | Rot.@15° | CC@0.1 |
| **RelPose++** | 11.0 | 57.0 | 28.6 | / | / | / | / | / | / |
| + SparseAGS | 27.3(+**16.3**) | 60.2(+**3.2**) | 47.0(+**18.4**) | / | / | / | / | / | / |
| **Ray Diff.** | 13.3 | 74.3 | 44.1 | 12.9 | 73.4 | 36.1 | 12.6 | 74.0 | 34.0 |
| + SparseAGS | 44.9(+**31.6**) | 83.6(+**9.3**) | 73.6(+**29.5**) | 70.0(+**57.1**) | 89.6(+**16.2**) | 83.4(+**47.3**) | 82.3(+**69.7**) | 93.2(+**19.2**) | 89.0(+**55.0**) |
| **DUSt3R** | 50.3 | 93.4 | 82.1 | 52.9 | 95.0 | 84.4 | 55.5 | 94.9 | 84.2 |
| + SparseAGS | 74.3(+**24.0**) | 95.1(+**1.7**) | 92.3(+**10.2**) | 87.0(+**34.1**) | 97.3(+**2.3**) | 94.3(+**9.9**) | 91.1(+**35.6**) | 97.7(+**2.8**) | 95.0(+**10.8**) |

## 4.2 Evaluation

**Camera Pose Accuracy**. We compare SparseAGS with SPARF [38] on pose accuracy given eight input images quantitatively in Tab. 1 (numbers are in percentage) and qualitatively in Fig. 3. We find that SparseAGS consistently yields larger improvements than SPARF, which sometimes even leads to degraded accuracy (marked by red numbers). We attribute this to the unreliable correspondences extracted by SPARF (we include an example in Fig. 7), as the input images in NAVI may exhibit more significant viewpoint changes compared to scene-level datasets, *e.g.*, DTU [10] where SPARF

Table 3: **Quantitative Comparison of 3D Reconstruction on NAVI [9].** We compare our method with two unposed approaches: LEAP [12] and UpFusion [13], using varying numbers of input images (N). We adopt two pose initializations (Ray Diff. [47], DUSt3R [41]) reporting PSNR and LPIPS.

| Method | Initial Cam. Pose | N = 6 | | N = 8 | | N = 10 | |
|---|---|---|---|---|---|---|---|
| | | PSNR ↑ | LPIPS ↓ | PSNR ↑ | LPIPS ↓ | PSNR ↑ | LPIPS ↓ |
| **LEAP** | ✗ | 12.84 | 0.2918 | 12.93 | 0.2902 | 12.98 | 0.2890 |
| **UpFusion** | ✗ | 13.30 | 0.2747 | 13.27 | 0.2744 | / | / |
| **SparseAGS** | Ray Diff. | 13.63 | 0.2697 | 15.30 | 0.2304 | 16.80 | 0.1960 |
| **SparseAGS** | DUSt3R | **15.56** | **0.2173** | **17.03** | **0.1870** | **18.03** | **0.1660** |

is originally tested. Note that training SPARF (or other NeRF-based methods) is far more expensive than ours, and it may take more than 10 hours. Whereas our method typically finishes in 5-10 minutes. More analysis and detailed comparisons with SPARF on pose accuracy and novel view synthesis are in Sec. D.

We vary the number of input images (N = 6, 10, 16) and report camera pose accuracy in Tab. 2 (we only test RelPose++ [18] with six images as inference with more than eight images is not supported). SparseAGS consistently enhances baseline performance for both rotation and camera center accuracy, with particularly significant gains for stricter metrics (*e.g.*, Rot.@5°). Moreover, the improvements tend to further increase with the number of input images. These results demonstrate that our method is robust to varying levels of initial camera poses and generalizes well across different input numbers.

**3D Reconstruction.** In Table 3, we compare SparseAGS with two unposed approaches, LEAP [12] and UpFusion [13], reporting metrics for 3D reconstruction (novel view synthesis). Our method consistently outperforms both baselines across different numbers of input images and with two pose estimation initializations. While SparseAGS shows continuous improvements with an increasing number of input images, the performance of LEAP and UpFusion nearly saturates in terms of both PSNR and LPIPS. We hypothesize that unposed methods struggle to utilize additional input images beyond their training capacity without further training adjustments (LEAP is trained using five views, while UpFusion is trained with a maximum of six images). In contrast, our method is flexible w.r.t. the number of input images, eliminating the need for re-training. A qualitative comparison with LEAP and UpFusion is presented in Fig. 4 and Fig. 5, respectively. The results show that SparseAGS better preserves the details in input images by explicitly modeling cameras and produces higher-quality novel view synthesis with more precise initial camera poses.

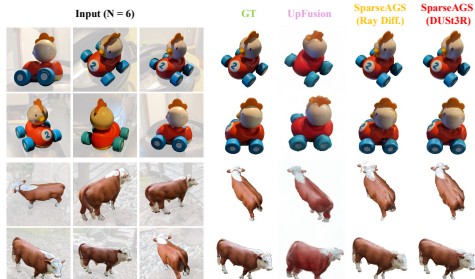

Figure 5: **Qualitative Comparison with UpFusion [13] on Novel View Synthesis.** We use two pose estimation baselines (Ray Diffusion [47] and DUSt3R [41]) as in Fig. 4. Note that the left eye and symbol ② of the Chicken Racer is missing in UpFusion's output, probably because of the "first-image bias", while SparseAGS preserves these details.

Table 4: **Ablation Study.** Using initial poses from Ray Diffusion [47] for eight input images, we ablate the effect of each proposed component of our approach.

| Method | | Rot.@5°↑ | Rot.@15°↑ | CC@0.1↑ | PSNR↑ | LPIPS↓ | F1@0.01↑ |
|---|---|---|---|---|---|---|---|
| Ray Diffusion | | 13.5 | 73.5 | 38.3 | / | / | / |
| + Pose-3D Co-opt. (w/o SDS) | (1) | 28.4 | 79.9 | 57.7 | 12.72 | 0.3100 | 46.3 |
| + SDS (vanilla Zero123 [20]) | (2) | 30.2 | 78.3 | 57.3 | 13.04 | 0.2999 | 49.9 |
| + SDS (Our 6-DoF Zero123) | (3) | 34.6 | 83.1 | 65.3 | 13.44 | 0.2793 | 57.2 |
| + Outlier Removal & Correction | (4) | 60.3 | 88.2 | 80.4 | 15.30 | 0.2304 | 68.2 |

## 4.3 Ablation Study

We ablate the effectiveness of each component in our approach (Tab. 4) using initial pose estimates from Ray Diffusion [47] with eight input images. In addition to camera pose metrics and novel-view

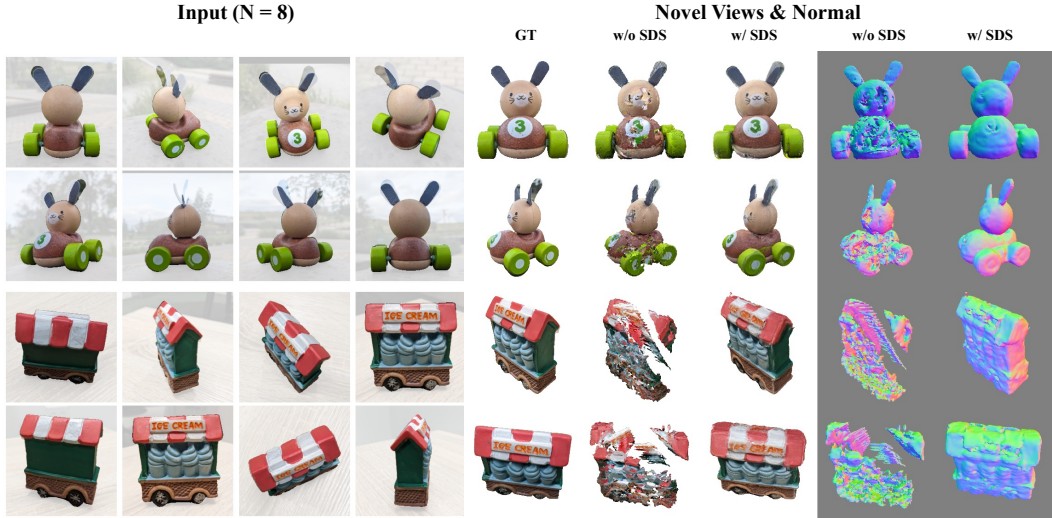

Figure 6: **Qualitative Comparison with No-Generative-Piror Setup (N = 8).** Novel Views (NV) & Normal: From left to right – GT, NV w/o SDS, NV w/ SDS, Normal w/o SDS, Normal w/ SDS. Leveraging generative priors in the form of SDS contributes to a consistent 3D representation.

metrics, we report the F1 score of reconstructed meshes, which reflects their alignment with ground truth meshes.

**(Appropriate) Generative Priors Improve Analysis by Synthsis**. We find that adding generative priors (Eq. 2) to naive pose-3D co-optimization (Eq. 1) improves both pose accuracy and 3D reconstruction quality (comparison between **(1)** and **(3)** shows consistent improvements in all metrics). However, vanilla Zero-1-to-3 [20] is not suitable for providing such priors in real-world scenarios, as we observed a drop in camera rotation and center accuracy (compare **(1)** with **(2)** in Rot.@15° and CC@0.1). This is because 3-DoF camera parameterization cannot well represent the camera poses in the wild. Although the numerical improvements may appear marginal (*e.g.*, in PSNR), Fig. 6 presents a qualitative comparison of 3D reconstruction with and without our 6-DoF novel-view generative priors. Supervision on novel views via SDS helps form a consistent 3D representation.

**Outlier Removal and Correction**. The presence of outlier initial cameras introduces significant challenges to pose-3D co-optimization. Our iterative outlier removal and correction pipeline effectively addresses this issue. For instance, comparing **(3)** with **(4)** shows a substantial improvement: Rot.@5° increased from 34.6% to 60.3% (**25.7**% absolute improvement), PSNR improved from 13.44 to 15.30, and F1@0.01 increased from 57.2 to 68.2 (**11.0** point absolute improvement). These results confirm the effectiveness of our approach.

# 5   Conclusion

In this work, we presented SparseAGS, a framework for joint pose estimation and 3D reconstruction – combining off-the-shelf pose estimation methods with a novel-view synthesis generative prior for robust inference in real-world sparse-view captures.

**Limitations**. While our experiments demonstrated clear improvements over initializations and stronger performance compared to prior 3D reconstruction methods, there are several challenges that remain. First, our approach does rely on some *reasonable* off-the-shelf pose estimates and cannot succeed if a large fraction of the predictions have a large error. Secondly, SparseAGS (similar to existing baselines) does not deal with truncation or occlusion and cannot be directly applied to scenarios with close-up images of parts of objects or cluttered scenes with one object occluding the other. Finally, we focused here on an object-centric setting, and it would be interesting to extend our approach to broader settings, *e.g.*, deploying our framework in conjunction with methods that learn novel-view generative priors for scenes.

## Acknowledgements

We thank Zihan Wang and the members of the Physical Perception Lab at CMU for their valuable discussions. We especially appreciate Amy Lin and Zhizhuo (Z) Zhou for their assistance in creating figures, as well as Yanbo Xu and Jason Zhang for their feedback on the draft.

This work was supported in part by NSF Award IIS-2345610. This work used Bridges-2 [1] at Pittsburgh Supercomputing Center through allocation CIS240166 from the Advanced Cyberinfrastructure Coordination Ecosystem: Services & Support (ACCESS) program, which is supported by National Science Foundation grants #2138259, #2138286, #2138307, #2137603, and #2138296.

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

## Appendix

## A  Broader Impacts

Our method leverages generative priors from diffusion models, enabling 3D reconstruction and pose estimation in the wild. This may benefit the generation of 3D assets for common users. However, we acknowledge that the web-scale data used for training these diffusion models may include content with potential negative social impacts, such as biased representations or harmful stereotypes. Therefore, while our approach benefits from the richness of the data, we must remain vigilant about the ethical implications and strive to mitigate any adverse effects.

## B  Analysis of Inference Time

For 8-image inference using a single RTX A5000 GPU, one reconstruction with MV-DreamGaussian takes about 2 minutes to complete, and the "render-and-compare" for each outlier takes around a minute. Our full pipeline (using RayDiffusion initialization) detected an average of 0.94 outliers per sequence on NAVI, resulting in an inference time of around 9 minutes. We believe additional engineering efforts can further optimize and reduce inference time.

## C  Implementation Details

**Finetuning Zero-1-to-3 with 6-DoF Camera Conditioning**. To learn the 6-DoF camera conditioning for novel view synthesis in the wild, we first initialize the weights of Zero-1-to-3 using the `Zero123-XL` checkpoint [5] and replace the original camera condition with ours. We then only finetune the layers associated with camera conditioning (*i.e.*, the linear projection and cross-attention layers) while freezing all other layers. This approach is more efficient than all-layer finetuning. To alleviate the synthetic data bias learned by the vanilla Zero-1-to-3, we include the training samples from CO3D [24] along with the Objaverse [6] renderings provided by Liu *et al.* [20] for finetuning. For computational resources, we used 8 V100 GPUs, setting a batch size of 36 per GPU with a gradient accumulation of 6. The model was trained for 23,500 iterations, taking ~4 days.

**Iterative Outlier Removal Details.** For the outlier condition specified in inequality 8, we employ LPIPS as the reprojection error metric, applying a threshold of 0.05. The reconstruction loop terminates when the average reprojection error reduction falls below this threshold or if the number of estimated inliers drops below a predefined count. Specifically, we use a threshold of 4 inliers for N = 6 and N = 8, 6 inliers for N = 10, and 12 inliers for N = 16. These iteration counts generally suffice to handle outliers given the current capabilities of state-of-the-art pose estimation systems.

**Comparing with Pose-free Methods.** To compare with pose-free methods, we follow these steps to obtain their novel view renderings: First, we normalize the ground truth camera poses to match the scale of the coordinate systems used by these methods. Next, we render target images from novel views using their relative camera poses with respect to the first input image. Additionally, we adjust the camera intrinsics (focal length and principal point) during inference to align the foreground mask of the rendered images with the ground truth to reduce scale difference.

## D  More Detailed Comparisons with SPARF

In addition to our main text comparing to SPARF for pose estimation, here we also present Novel View Synthesis (NVS) metrics. We report the results in Tab. 5, using DUSt3R pose (N = 6, 8) as initialization. Due to SPARF's long training time (about 10 hours per instance), we could only include

Table 5: **Expanded Comparison of Pose Accuracy and Novel View Synthesis with SPARF [38].** In addition to the primary metrics presented in the main text, we report Average Rotation Error and Improvement Rate (IR), which indicates the percentage of sequences with reduced pose error. See Sec. D for further analysis and detailed explanations.

| N = 6 | Pose Metrics | | | | NVS Metrics | | | |
|---|---|---|---|---|---|---|---|---|
| | Avg Rot. Err | Rot.@5° | CC@0.1 | IR | PSNR | LPIPS | *PSNR | *LPIPS |
| DUSt3R | 7.90 | 48.5 | 80.2 | / | / | / | / | / |
| w/ SPARF | 17.07 | 47.9 | 67.6 | 0.41 | 12.80 | 0.3201 | 15.30 | 0.2478 |
| w/ SparseAGS | 5.82 | 73.9 | 92.1 | 0.80 | 15.52 | 0.2179 | 16.23 | 0.1844 |
| N = 8 | Pose Metrics | | | | NVS Metrics | | | |
| | Avg Rot. Err | Rot.@5° | CC@0.1 | IR | PSNR | LPIPS | *PSNR | *LPIPS |
| DUSt3R | 8.71 | 51.6 | 79.5 | / | / | / | / | / |
| w/ SPARF | 18.17 | 57.3 | 68.6 | 0.54 | 13.42 | 0.3059 | 15.44 | 0.2584 |
| w/ SparseAGS | 5.99 | 81.7 | 92.5 | 0.93 | 17.02 | 0.1874 | 17.48 | 0.1695 |

70 sequences (2 sequences per object) in these two experiments. Notably, for a direct comparison on NVS, we removed backgrounds from the input images, whereas no masking was applied in Tab. 1. This may slightly disadvantage SPARF, as backgrounds provide additional cues for pose registration and correspondences.

The results indicate that SparseAGS outperforms SPARF in pose accuracy and novel view quality as well. In fact, we found that SPARF can often make the poses worse compared to the (relatively accurate) DUSt3R initialization (measured via *Improvement Rate* that indicates the percentage of sequences with reduced pose error). This is likely because the correspondences leveraged by SPARF in its optimization are not robust and are susceptible to false matches – see Fig. 7 for an example.

As an attempt to compare novel view quality despite the difference in pose accuracy, we report *PSNR and *LPIPS, which are measured *only on the sequences where SPARF improves pose accuracy* and find that even in these, our approach outperforms it. We also observed that while SPARF works well on novel views close to the input, floaters constantly appear with significant viewpoint changes. In contrast, our generative prior leads to a more consistent 3D representation.

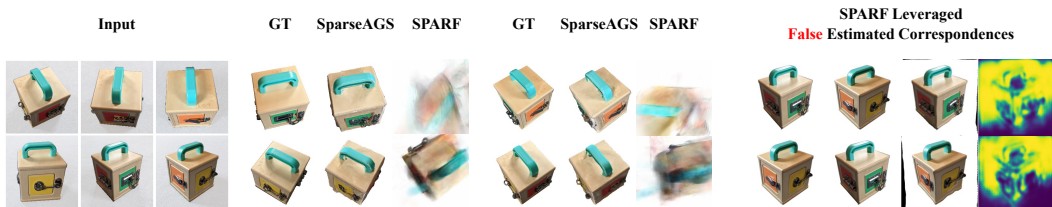

Figure 7: **SPARF Fails When Incorrect Correspondence is Leveraged.** Rightmost Section: From left to right – Source, Target, Warped Source to Target Based on Estimated Correspondence, Confidence Map (yellow indicates high confidence). The estimated false correspondence due to symmetric patterns causes pose optimization to fail, leading to degraded novel views in SPARF.

# E   Evaluation on Synthetic Datasets

Table 6: **Evaluation of Rotation Accuracy on Three Synthetic Datasets (GSO [7], ABO [4] and OmniObject3D [44]).** We test our method on ID-Pose [3] with eight images as input. We measure rotation accuracy at two thresholds (15 and 30 degrees).

| Method | GSO | | ABO | | OmniObject3D | |
|---|---|---|---|---|---|---|
| | Rot.@15° | Rot.@30° | Rot.@15° | Rot.@30° | Rot.@15° | Rot.@30° |
| ID-Pose | 52.5 | 59.2 | 47.3 | 52.7 | 55.1 | 62.4 |
| w/ SparseAGS | 68.8(**+16.3**) | 74.8(**+15.6**) | 64.4(**+17.1**) | 69.0(**+16.3**) | 75.1(**+20.0**) | 79.2(**+16.8**) |

Though our main focus is real-world data, we apply our approach to three synthetic datasets (GSO, ABO, OmniObject3D) for a complete evaluation of our approach. We use ID-Pose [3] as a baseline,

which inverses the novel-view-synthesis ability of Zero-1-to-3 [20] for pose estimation and adopts a 3-DoF camera parameterization. Here, we also use the vanilla Zero-1-to-3 for multi-view SDS loss to accommodate this camera definition and for fair comparison. We report the results on camera rotation accuracy in Tab. 6. Across these datasets, our approach consistently improves performance on two metrics, even though ID-Pose uses the same backbone model (vanilla Zero-1-to-3) as we do. These results demonstrate that our approach more effectively leverages the generative priors from Zero-1-to-3, achieving better pose accuracy. Plus, we also show that our method is applicable to synthetic data, showing strong generalization abilities across different datasets.

## F    Additional Visualizations for Inferred Poses

See Fig. 8.

## G    Additional Visualizations for Novel View Synthesis

See Fig. 9.

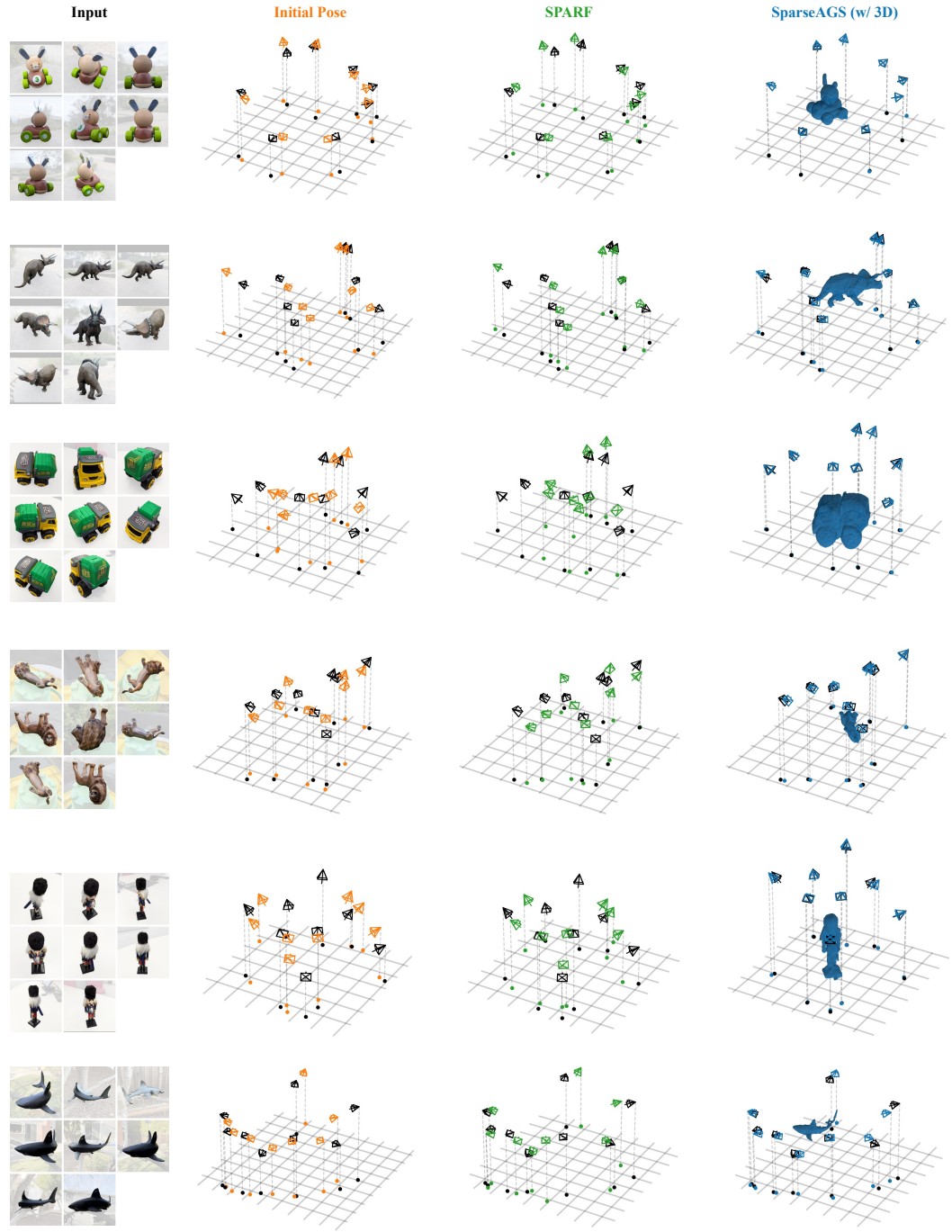

Figure 8: **More Qualitative Comparison on Camera Pose Accuracy.** Given initial camera poses from off-the-shelf methods, the refined poses from SPARF [38] are compared with the output of SparseAGS. We align the estimated cameras with ground truth (in black) with an optimal similarity transform.

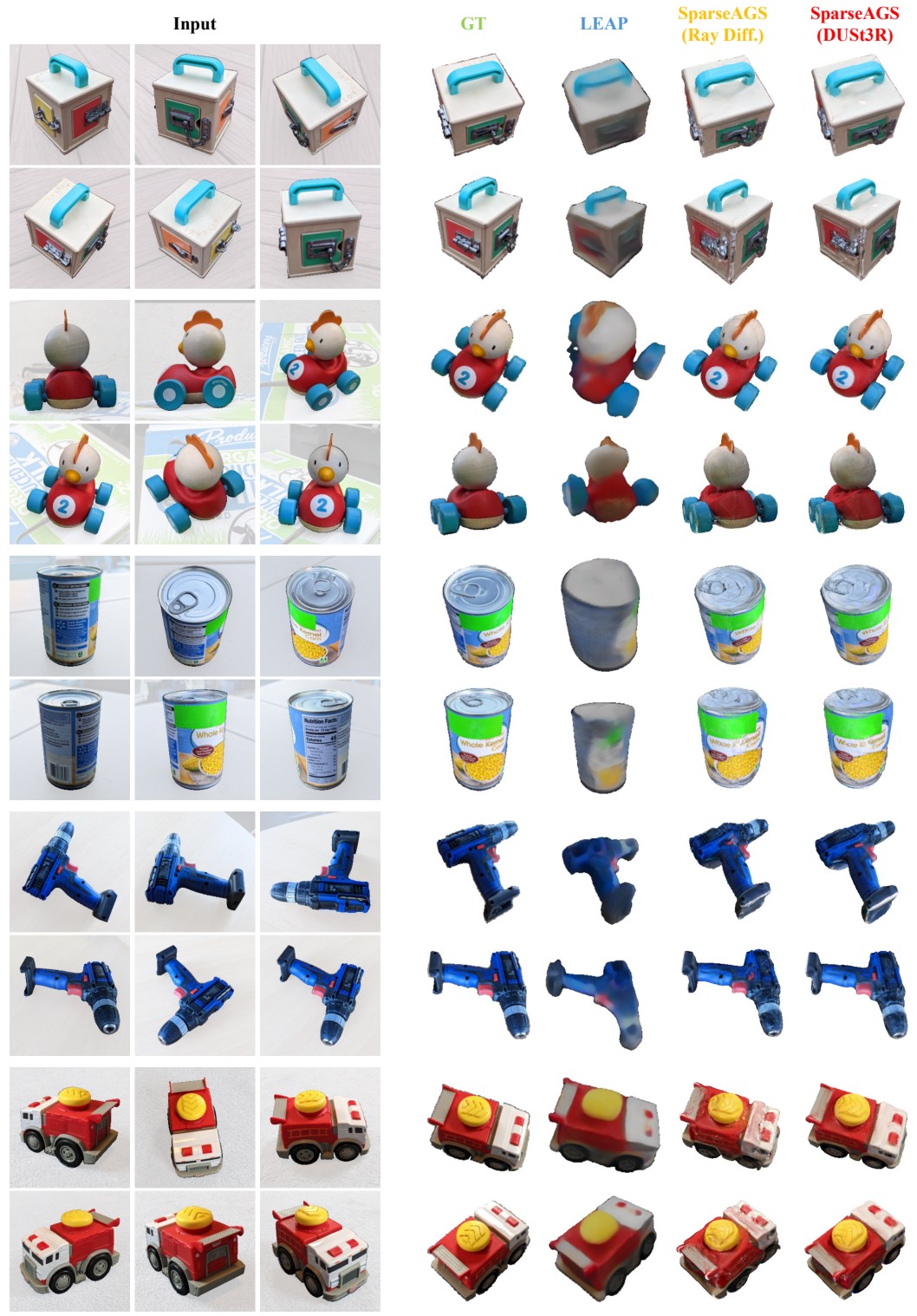

Figure 9: **More Qualitative Comparison with LEAP [12] on Novel View Synthesis.** We use two pose estimation baselines (Ray Diffusion [47] and DUSt3R [41]). SparseAGS better preserves details in the input images and shows enhanced performance with more accurate initial camera poses.

