# OpenReview forum: "Sparse-view Pose Estimation and Reconstruction via Analysis by Generative Synthesis"
_NeurIPS.cc/2024/Conference — NeurIPS 2024 poster_

### Official Review · Reviewer_XaBT · 2024-07-09

**Soundness:** 3
**Presentation:** 3
**Contribution:** 3
**Rating:** 6
**Confidence:** 5

**Summary:**

The paper introduces a novel optimization-based method for sparse-view 3D reconstruction from unposed images. The method uses off-the-shelf pose estimator to get pose initialization, then it uses rendering loss and generative priors to optimize the pose and 3D reconstruction. In detail, the generative priors involve a multi-view SDS loss on generated novel views using Zero123. The method demonstrates satisfying results on the evaluation data, and the ablation study shows the effectiveness of each proposed technique.

**Strengths:**

- Good performance. The reconstruction quality and pose estimation accuracy are satisfying.
- The paper is well-written and is easy to follow.
- The idea of rejecting images with large pose error is interesting.
- The technical part of the paper is solid.

**Weaknesses:**

- Missing baseline. For the reconstruction methods, the only baseline is LEAP, which is a feedforward method. In contrast, the proposed method is an optimization-based method, which introduces pose-processing to estimated poses. I would suggest adding baseline of SPARF [1] and using the same pose initialization. Moreover, why not comparing with UpFusion?
- Unknown inference speed. Will the joint optimization of pose and shape be slow? Could you provide a analysis of inference time?
- Related work. One related work is iFusion [2], which uses generative priors for pose estimation and is very relevant to the philosophy of the proposed method. Another related work is FORGE [3], which introduces pose optimization for sparse view reconstruction. Moreover, the authors should discuss the prior sparse-view reconstruction from unposed images works with more details, the authors should provide more comparison and contrast with prior work. The current discussion is too short (Line 90-92).
- Ablation study. The ablation study is performed with the Ray Diffusion pose initialization. How will it look like using Dust3r initialization? This is important as the ablation should be performed with the best base model.


[1] Truong, Prune, et al. "Sparf: Neural radiance fields from sparse and noisy poses." CVPR 2023.
[2] Wu, Chin-Hsuan et al. “iFusion: Inverting Diffusion for Pose-Free Reconstruction from Sparse Views.” ArXiv 2023.
[3] Jiang, Hanwen et al. “Few-View Object Reconstruction with Unknown Categories and Camera Poses.” 3DV 2024.

**Questions:**

- The introduction spends a lot space discussing the chicken-and-egg problem of pose estimation and reconstruction. However, I don't think it is quite related to the technical part, as the proposed method still need pose initialization using off-the-shelf methods. The method doesn't provide a novel perspective regarding how to solve the chicken-and-egg problem, and using pose initialization is quite common in prior works, e.g., SPARF, FORGE, and FvOR or even traditional SfM methods. Why the authors want to emphasize this?
- Is it possible to evaluate the outlier removal method? For example, the authors can evaluate the correlation between the removal and the pose error. If the proposed method works well, they should have strong correlations. Moreover, it will be good to provide any statistics on the outlier removal method, e.g., how many images are removed in average.

**Limitations:**

Please see weaknesses and questions.

---

> ### Author Rebuttal · Authors · 2024-08-06
>
> > Missing baseline.
>
> We would like to thank the reviewer for the suggestion on baselines. We include the comparison between the proposed approach and SPARF in General Response. In additions, we also compare our method with UpFusion on novel view synthesis:
>
> | Method   | Init Cam Pose | (N=6) PSNR | LPIPS  | (N=8) PSNR | LPIPS  |
> | :------- | :------------ | :--------- | :----- | :--------- | :----- |
> | LEAP     | /             | 12.84      | 0.2918 | 12.93      | 0.2902 |
> | UpFusion | /             | 13.30      | 0.2747 | 13.27      | 0.2744 |
> | Ours     | Ray Diffusion | 13.63      | 0.2697 | 15.30      | 0.2304 |
> | Ours     | DUSt3R        | 15.56      | 0.2173 | 17.03      | 0.1870 |
>
> UpFusion performs slightly better than LEAP, while our approach constantly outperforms both pose-free baselines with two different pose initialization and view numbers. Our method improves consistently by leveraging more precise initial camera poses and more observed input images. However, LEAP and UpFusion show little performance increase when more views are available due to their geometry unawareness. We also include a qualitative comparison between our approach and UpFusion in **Fig. 4** of the rebuttal PDF file.
>
> > Unknown inference speed.
>
> We apologize for the ambiguity in our earlier version. Our shape-pose optimization is fast, benefiting from the efficient Gaussian Splatting compared to NeRF baselines, and on average our inference time is about 9 minutes, whereas SPARF may take more than 10 hours to train a full model. We discuss inference time more thoroughly in our General Response and will revise our paper according to the review's feedback.
>
> > Related work.
>
> Thanks for mentioning related work! We will incorporate our discussion into our updated version.
>
> + iFusion leverages diffusion priors from the diffusion model (i.e., Zero-123) for pose estimation in a similar manner to ID-Pose where relative camera poses are parameterized as conditioning to the diffusion model and are refined via gradient descent from diffusion noise prediction. Compared to iFusion, we also leverage diffusion priors from Zero-123 but we optimize camera poses directly via photometric error from differentiable rendering. In this process, we explicitly form a 3D representation (3DGS) for optimization.
> + FORGE (which is earlier work from the authors of LEAP) uses the synergy of pose estimation and 3D reconstruction by inferring 3D features that are shared by both tasks. Our work shares a similar high-level idea in that we make use of the estimated camera poses for 3D reconstruction, and the reconstructed 3D, in turn, is exploited to help refine camera poses and identify outliers. By doing this, we also hope to make both sub-tasks to benefit from each other.
>
> In our final version, we will add detailed discussions about pose-free approaches (e.g., LEAP, UpFusion) in our Related Work section.
>
> > Ablation study.
>
> We thank the reviewer for suggesting an ablation study on DUSt3R, and present the results below:
>
> |                                 | **Rot@5°** | **Rot@15°** | **CC@10%** | PSNR  | LPIPS  | **F1** |
> | ------------------------------- | ---------- | ----------- | ---------- | ----- | ------ | ----------- |
> | DUSt3R                          | 52.3       | 93.8        | 82.2       | /     | /      |             |
> | \+ Pose-3D Co-opt. (w/o SDS)    | 80.9       | 95.8        | 91.6       | 16.79 | 0.2072 | 64.70       |
> | \+ SDS (vanilla Zero123)        | 78.7       | 95.3        | 90.2       | 16.38 | 0.2097 | 69.45       |
> | \+ SDS (Our 6-DoF Zero123)      | 81.1       | 95.9        | 91.0       | 16.75 | 0.1941 | 75.04       |
> | \+ Outlier Removal & Correction | 83.7       | 96.2        | 93.5       | 17.03 | 0.1870 | 75.80       |
>
> > The introduction spends a lot space discussing the chicken-and-egg problem of pose estimation and reconstruction... Why the authors want to emphasize this?
>
> We agree with the reviewer that classical SfM methods did indeed tackle this chicken-and-egg problem. Our introduction was instead motivated by ignorance of the chicken-and-egg nature in the more recent learning-based sparse-view reconstruction methods. For example, approaches like LEAP/UpFusion seek to side-step the pose estimation task altogether. On the other hand, methods that explore generative-prior-based reconstruction (e.g. ReconFusion/SparseFusion) assume perfect poses. Even methods that jointly optimize 3D and pose (e.g. BARF/SPARF/FvOR) are designed to use noisy ground-truth poses or near-perfect initial poses (as opposed to ones output by real systems with possibly large errors).
>
> We agree this emphasis maybe obvious to any reader familiar with the rich history of the field, but believe this is useful for readers only familiar with a recent history.  That said, we would be happy to revise the introduction if the reviewer suggests.
>
> > Statistics of Outlier Removal.
>
> We again thank the reviewer for this advice. With the same setting as Table 4 in our main paper, we investigated the relationship between the identified outlier numbers and the corresponding sequence numbers in our dataset (174 in total) and their initial average rotation error.
>
> | # of Outliers            | 0     | 1     | 2     | 3     | 4     |
> |--------------------------|-------|-------|-------|-------|-------|
> | Init Rot Error | 14.19 | 13.09 | 20.52 | 17.16 | 33.17 |
> |   # of Sequences         | 56    | 82    | 28    | 7     | 1     |
>
> In general, sequences with higher rotation errors tend to be identified with more outliers. In this setup, our method identified ~0.94 outliers per sequence. We also computed the rotation error at an image level for outliers and inliers with the average over all samples as reference:
>
> All: 14.92°
> Outlier: 19.75°
> Inlier: 13.53°
>
> The results indicate that the outlier we found indeed has a higher error than others, verifying the effectiveness of our outlier identification approach.

---

> > ### Comment · Reviewer_XaBT · 2024-08-08
> >
> > Thanks for the detailed reply! The rebuttal resolves most of my concerns. I would like to raise the score of this paper.
> >
> > Before that, I have a last question. I noticed that in the reply to Reviewer mujy, with using SPARF, the pose error becomes larger, which is counter-intuitive. Could the authors provide more insight into this? Is there any problem with the estimated correspondence, e.g. does the object-centric image makes the correspondence fail? If so, is it reasonable to use images with original background as the input of SPARF, and after rendering the novel views, we use foreground segmentation to evaluate the NVS metrics?

---

> > > ### Author Response · Authors · 2024-08-09
> > >
> > > We would like to thank the reviewer for the prompt response and the willingness to raise our score!
> > >
> > > Yes, we indeed found that correspondence is crucial for SPARF's pose optimization, and we observed that SPARF often fails to obtain reliable correspondence in our experiments. In **Fig. 3** of our rebuttal PDF file, we included an example where false estimated correspondence makes SPARF's pose estimation fail, and therefore, it produced degraded novel views.
> > >
> > > To demonstrate the influence of foreground masks on correspondence (which is reflected by pose accuracy), we present a comparison of SPARF with and without using masks below.
> > >
> > > | N=8              | Avg Rot Error | Rot@5° | Rot@15° | CC@10% | Improvement Rate |
> > > | :--------------- | :------------ | :----- | :------ | :----- | :--------------- |
> > > | DUSt3R           | 8.71          | 51.6   | 92.9    | 79.5   | /                |
> > > | SPARF (w/ mask)  | 18.17         | 57.3   | 75.7    | 68.6   | 0.54             |
> > > | SPARF (w/o mask) | 10.10         | 58.1   | 85.9    | 81.2   | 0.59             |
> > > | Ours             | 5.99          | 81.7   | 95.3    | 92.5   | 0.93             |
> > >
> > > While including background did improve SPARF's pose accuracy, it failed to improve the average rotation error and Rot@15° over the baseline as the w/ mask setting did. Even in this setup, we still observed false estimated correspondence used by SPARF, which may cause large pose errors. Given these results, we may attribute the SPARF's counter-intuitive performance more to the distribution of NAVI data instead of the use of masks. In NAVI, images have limited overlap, making it inherently hard to find reliable correspondence. This differs significantly from SPARF's testing data in the original paper, which may show notable overlaps and a strong forward-facing feature (e.g., the DTU dataset).
> > >
> > > In our General Response, we also tried to compare novel view synthesis with SPARF despite differences in pose accuracy by introducing `*PSNR` and `*LPIPS`.  We used these metrics to factor out the negative influence of failure pose optimization on SPARF due to unavailable/false correspondence. We refer the reviewer to our General Response for more details! We hope this could address the reviewer's concern regarding the influence of masks (and thus the influence of pose accuracy) on NVS.
> > >
> > > Finally, we would like to clarify an implementation detail relevant to the reviewer’s question: in the original submission, when reporting pose error, we did give SPARF access to the background as it gave stronger results. Unfortunately, for NVS experiments, this was not trivial as it would require post-processing to remove background from the renderings before comparing NVS metrics, so we gave it images w/o background. Although this may be slightly suboptimal, we hope that our `*PSNR` and `*LPIPS` (which explicitly focus on sequences where SPARF improved pose, thus minimizing the effect of sequences where SPARF was not robust) make it clear that our system does outperform SPARF.

---

> > > > ### Comment · Reviewer_XaBT · 2024-08-09
> > > >
> > > > Thanks for the reply. The response makes sense to me. I will raise my score.

---

> > > > > ### Author Response · Authors · 2024-08-10
> > > > >
> > > > > We are glad to see that the reviewer's concerns were resolved, and we will update our paper based on our discussion with the reviewer.

---

### Official Review · Reviewer_mujy · 2024-07-10

**Soundness:** 3
**Presentation:** 2
**Contribution:** 3
**Rating:** 6
**Confidence:** 4

**Summary:**

This paper proposes a framework for joint 3D reconstruction and pose refinement. Specifically, given estimated camera poses from off-the-shelf models, the proposed method first leverages diffusion priors and rendering loss for 3D reconstruction. The 3D reconstruction is further used to refine the current pose parameters. The 3D reconstruction and pose refinement are conducted in an alternative way. An outlier identification and correction strategy is also introduced to make full use of the given image while mitigating the adverse effect of noisy camera estimations at the same time. Experimental comparison with several pose estimation baselines shows that the proposed method can refine inaccurate pose estimation effectively.

**Strengths:**

1. The paper tackles a practical problem in real-world scenarios, where ground truth camera poses are not always available.
2. The proposed method is shown to be effective when applying to different pose estimation baselines.
3. The proposed outlier removal and correction is effective from the ablation study results in Table 4.

**Weaknesses:**

1. The proposed method is compared with SPARF only in the setting of using pose from different pose estimation baselines. However, it would be more convincing to also present the results using the same setting of SPARF, which adds noise into the GT camera pose. This will be a direct comparison with SPARF’s original results reported in their paper.
2. The proposed method is compared with LEAP for 3D reconstruction results. However, the comparison is a bit unfair since LEAP does not require any initial camera poses.
3. The description of how to effectively detect the outliers (line 212 - line 214) is not very clear. Similarly, the procedure of how to correct the outlier poses (line 223 - line 225) is not very clear either. How the MSE and LPIPS are computed and compared since there is no correspondence?

**Questions:**

1. The proposed method is evaluated on the NAVI dataset. It seems that the dataset is quite simple as shown in Fig. 3 and Fig. 4. The reviewer is wondering about the performance of the proposed method on more complex scenes?
2. The reviewer is wondering about the separate ablation results on the outlier removal and correction.

**Limitations:**

Limitations are addressed in the paper.

---

> ### Author Rebuttal · Authors · 2024-08-06
>
> > The proposed method is compared with SPARF only in the setting of using pose from different pose estimation baselines. However, it would be more convincing to also present the results using the same setting of SPARF, which adds noise into the GT camera pose. This will be a direct comparison with SPARF’s original results reported in their paper.
>
> Thanks for the advice! We experimented with the suggested setting using eight images, where Gaussian noise is added to ground truth camera poses (so the initial poses have a rotation error of about 9.71°). A comparison between the proposed method and SPARF on  pose accuracy is presented below:
>
> | Method         | Avg Rot Error | Rot@5° | Rot@15° | CC@10% | Improvement  Rate |
> | :------------- | :------------ | :----- | :------ | :----- | :------------ |
> | Initialization | 9.71          | 11.9   | 89.1    | 56.3   | /             |
> | w/ SPARF       | 14.47         | 43.6   | 73.5    | 66.5   | 0.53          |
> | w/ Ours        | 3.63          | 82.4   | 96.9    | 94.4   | 0.92          |
>
> While our method yields consistent improvements over all metrics, we observed that SPARF can sometimes degrade performance compared to the initialization.  As SPARF leverages correspondence, its pose optimization is ineffective (or even diverges) when no reliable correspondence (or false correspondence) is found. The results are consistent with our experiments on DUSt3R poses, which we include in our General Response. We refer the reviewer to the top-level response for more details. Please note that the experiments in the original SPARF paper were done using image sets with significant overlap (e.g.  DTU dataset where all images observe the same aspect of an object).
>
> > The proposed method is compared with LEAP for 3D reconstruction results. However, the comparison is a bit unfair since LEAP does not require any initial camera poses.
>
> For an additional baseline, please see our general response where we additionally compare to SPARF on novel-view synthesis. We would also be happy to report any additional comparisons the reviewer suggests.
>
> > The description of how to effectively detect the outliers (line 212 - line 214) is not very clear. Similarly, the procedure of how to correct the outlier poses (line 223 - line 225) is not very clear either. How the MSE and LPIPS are computed and compared since there is no correspondence?
>
> We apologize for being unclear in our submission and will update the text based on the reviewer's feedback. In outlier correction, we do render-and-compare for each identified outlier, where we resue the reconstructed 3D from only the inliers (obtained in *Iterative Outlier Identification*). By sampling pose candidates, we can render images that can be compared with the target image (of the outlier) with metrics such as MSE and LPIPS. Therefore, correspondence is not required in the process. We included detailed explanations for outlier removal and correction in our General Response, and we hope that this will resolve the reviewer's concern.
>
> > The proposed method is evaluated on the NAVI dataset. It seems that the dataset is quite simple as shown in Fig. 3 and Fig. 4. The reviewer is wondering about the performance of the proposed method on more complex scenes?
>
> We primarily tested on NAVI because it provides ground truth 3D meshes that are unavailable in other datasets. This enables the evaluation of 3D metrics such as F1 for our 3D reconstruction. In addition, NAVI offers highly accurate object masks and high-quality sparse-view image sequences. However, we include qualitative results of our approach on more challenging scenarios in **Fig. 1** of our rebuttal PDF, where we show two self-captures and four challenging instances from the held-out set of CO3D. These instances have more complex textures and shapes than the NAVI samples, which verifies the proposed approach's generalization ability.
>
> > The reviewer is wondering about the separate ablation results on the outlier removal and correction.
>
> We would like to thank the reviewer for mentioning this ablation. We followed our setup in Table 4 of the main paper, but we only do *Iterative Outlier Identification* without the following correction stage.
>
> | Method                          |   Rot@5°  |   Rot@15°  | CC@10%  | PSNR  | LPIPS  | F1    |
> |---------------------------------|-----------|------------|---------|-------|--------|-------|
> |   Ray Diffusion                 |   13.5    | 73.5       | 38.3    | /     |   /    |   /   |
> |   Ours (w/o Outlier Correction) | 47.8      | 86.6       | 75.2    | 14.45 | 0.2494 | 65.41 |
> | Ours (Full System)              | 60.3      | 88.2       | 80.4    | 15.30 | 0.2304 | 68.19 |
>
> Compared to the baseline Ray Diffusion numbers, this experiment setup benefits from iterative pose-3D co-optimization, improving the base pose accuracy by a notable margin. Moreover, the results verify that an additional outlier correction is necessary for higher pose accuracy and high-fidelity novel views.

---

> > ### Comment · Reviewer_mujy · 2024-08-13
> >
> > Thanks for the authors' detailed responses. Most of my concerns are resolved in the rebuttal. For the comparison on 3D reconstruction, it would be more convincing to compare under the same condition, namely based on an initial pose. However, LEAP is not using pose information, hence putting LEAF at disadvantage for the comparison.

---

> > > ### Author Response · Authors · 2024-08-13
> > >
> > > Dear Reviewer,
> > >
> > > Thanks for the feedback. We believe there are two schools of thought for developing approaches for sparse-view-3D in-the-wild: a) methods that side-step explicit pose estimation (e.g. LEAP, UpFusion), and b) methods that rely on and improve initial pose estimation (e.g. SPARF and ours). We do already include a comparison to SPARF as a baseline that also uses the same initial poses to show that for methods following ideology (b), our work improves over the current SOTA.
> > >
> > >
> > > Regarding the comparison to LEAP, we certainly agree that LEAP not using poses is perhaps a reason why it is not competitive -- but this is exactly the point we wish to make! In particular, we hope that one key take-away for a reader is that methods following ideology (a) are limited in their performance. Based on their comments, this maybe something the reviewer feels is obvious, but this is not universally agreed! For example, LEAP positions itself as ‘liberating’ sparse-view 3D reconstruction from a dependence on pose estimation and even begins the abstract with a question that “Are camera poses necessary for multi-view 3D modeling?” and argues that they are not needed (UpFusion also follows a similar philosophy). Our comparison to LEAP (and the added comparison to UpFusion in response to Reviewer XaBT) seeks to make the counterpoint that “Actually, such poses are indeed helpful!”.
> > >
> > > We would be happy to present the results with this context more clearly outlined in the text if the reviewer feels that maybe helpful.

---

### Official Review · Reviewer_uJqP · 2024-07-11

**Soundness:** 3
**Presentation:** 3
**Contribution:** 2
**Rating:** 6
**Confidence:** 4

**Summary:**

This paper proposes a method for the joint reconstruction of camera poses and 3D objects given sparse input views. The core idea is to use a pose-conditioned diffusion model (Zero-123) as a prior, impose the SDS loss, and jointly optimize the poses and objects, similar to the approach in ID-pose. To improve the robustness and quality of the optimization, the authors made several modifications: (1) Using a 6 DoF pose-conditioned diffusion model instead of a 3 DoF model. (2) Adding strategies for outlier detection and correction. (Although somewhat empirical, it proves effective.)

This approach requires initial camera poses (from methods such as RelPose++, RayDiffusion, etc.) and is not capable of reconstructing poses from scratch (e.g., purely random camera poses). Experimental results demonstrate that, compared to SPARF and ID-pose, the proposed method achieves better pose estimation quality. Additionally, it provides better object reconstruction in terms of novel view synthesis quality compared to LEAP.

**Strengths:**

(1) The approach is technically sound, and I believe the reported results are reproducible.

(2) The reconstructed results look good and represent the state-of-the-art in object-level pose-free reconstruction.

(3) The paper is well-written, making it easy to read and understand.

**Weaknesses:**

(1) This optimization-based method requires more time compared to a feed-forward model, taking about 5-10 minutes. Additionally, the writing discussing this aspect is somewhat unclear: the paper states, “with increased inference time depending on the number of outliers.” Could this statement be more specific? How much does the time increase with the number of outliers? The correction of outliers may be time-consuming as it requires dense searches of initial camera poses.

(2) (Minor) The method focuses only on object-level reconstruction, which makes the scope seem narrow.

(3) The authors do not sufficiently discuss experiments in a more “standard” sparse-view setting, such as using 3 or 4 views. The reported experiments use at least 6 views, which is not a particularly small number.

**Questions:**

(1) A related work is lacking in discussion:  Sun, Yujing, et al. "Extreme Two-View Geometry From Object Poses with Diffusion Models." arXiv preprint arXiv:2402.02800 (2024).

(2) Is the testing data included in the training set for fine-tuning the 6-DoF diffusion model?

**Limitations:**

As discussed in the weakness.

---

> ### Author Rebuttal · Authors · 2024-08-06
>
> > This optimization-based method requires more time compared to a feed-forward model, taking about 5-10 minutes. Additionally, the writing discussing this aspect is somewhat unclear: the paper states, “with increased inference time depending on the number of outliers.” Could this statement be more specific? How much does the time increase with the number of outliers? The correction of outliers may be time-consuming as it requires dense searches of initial camera poses.
>
> We apologize for the lack of clarity and address the inference time in our general response above, and will update the final version to include these details. We agree that our system is slower than feed-forward methods (e.g LEAP), but allows significantly more accurate generations. Compared to prior NeRF-based optimization methods however, our gaussian-splatting based system is far more efficient e.g. SPARF takes 10hrs per instance whereas our systems takes ~9min.
>
> > The authors do not sufficiently discuss experiments in a more “standard” sparse-view setting, such as using 3 or 4 views. The reported experiments use at least 6 views, which is not a particularly small number.
>
> We chose 6-8 images as a setting representative of online marketplaces, but agree with the reviewer that it is important to also study even fewer views and report an experiment analyzing N=4 below.
>
> | Method  | Rot@5° | Rot@15° | CC@10% |
> | ------- | :----: | :-----: | :----: |
> | DUSt3R  |  52.0  |  90.3   |  79.0  |
> | w/ Ours |  57.8  |  90.5   |  84.5  |
>
> While our system does lead to consistently improved poses, the gains are less prominent compared to settings with more views. We believe this is because our approach does rely on multiple images 'co-observing' common 3D regions (to guide pose correction), but these are not common if we randomly sample a small set of views around an object, thus leading to diminishing benefits with very few images.
>
> > A related work is lacking in discussion: Sun, Yujing, et al. "Extreme Two-View Geometry From Object Poses with Diffusion Models." arXiv preprint arXiv:2402.02800 (2024).
>
> We thank the reviewer for sharing relevant work, which we will incorporate in our final version. The process of matching the generated images with the target image is similar to our outlier correction procedure at a high level, but we leverage a reconstructed 3D representation to render novel views instead of purely using diffusion models to generate them. Regarding this perspective, our novel views may be more detailed and faithful as we leverage multiple input views.
>
> >  Is the testing data included in the training set for fine-tuning the 6-DoF diffusion model?
>
> No, we did **not** include NAVI in our data to fine-tune Zero-123 with 6-DoF camera conditions.
>
> > The core idea is to use a pose-conditioned diffusion model (Zero-123) as a prior, impose the SDS loss, and jointly optimize the poses and objects, similar to the approach in ID-pose.
>
> We would like to clarify that our method is not similar to ID-Pose. ID-Pose estimates camera poses by optimizing the relative pose conditions in Zero-123 for image pairs while it does not form any 3D representation. In fact, we adopted ID-Pose as initialization for our experiments on three synthetic datasets in the supplementary and show that our systems leads to significant gains over it.

---

### Official Review · Reviewer_ePgG · 2024-07-14

**Soundness:** 3
**Presentation:** 3
**Contribution:** 2
**Rating:** 6
**Confidence:** 3

**Summary:**

This paper presents a method named MV-DreamGaussian for tackling the problem of 3D reconstruction from sparse multi-view inputs. In particular, the paper extends the DreamGaussian work to use multi-view images as the inputs and proposes a scheme to optimize the inaccurate camera poses of the multi-view images.

**Strengths:**

- This paper is well written and I can follow smoothly.
- The authors proposed a finetuned version of Zero-1-to-3 with 6 DoF camera parametrization which shows an advantage over 3 DoF camera parameterization in the original paper.
- The proposed pose refinement scheme is novel and very effective according to the authors' experiments compared with SPARF as well as the ablation study which shows that adding the proposed pose refinement improves the pose accuracy and reconstruction quality significantly. The design of the outlier removal based on photometric error ranking and discrete search is empirical but works quite well.

**Weaknesses:**

- This paper presents very limited novelty in the reconstruction part with a trivial extension to DreamGaussian to use multi-view images, which is already implemented in a public repository [stable-dreamfusion](https://github.com/ashawkey/stable-dreamfusion).
- The major weakness of the paper is the lack of fair comparisons in terms of the 3D reconstruction. The authors only compared with LEAP for the 3D reconstruction. However, LEAP is a work that **does not require any pose inputs**, whereas the proposed work needs relatively good pose initialization (e.g., Dust3r) and conduct refinement on it. In addition, the underlying 3D representation is different, too: LEAP uses NeRF while the proposed work uses 3D Gaussian. I'm confused as to why the authors did not compare with SPARF for the reconstruction quality too since SPARF shares the same input setup as the proposed work. Besides, the very recent work DMV3D would also be a good method to compare with.

**Questions:**

- I'm quite curious what the reconstruction quality the method can achieve without 3D generative prior but with the proposed refinement. Namely the combination of (1) and (4) in Table 4.
- How are the poses used for generative prior sampled in addition to the input views?
- How are the thresholds for pose outlier removal tuned?

**Limitations:**

The authors have discussed the limitations of the paper and I generally agree with them.

---

> ### Author Rebuttal · Authors · 2024-08-04
>
> > This paper presents very limited novelty in the reconstruction part with a trivial extension to DreamGaussian to use multi-view images, which is already implemented in a public repository stable-dreamfusion.
>
> We respectfully disagree with the reviewer's assessment that our extension to DreamGaussian is trivial. While using multiple input images for DreamGaussian has been implemented in the stable-dreamfusion GitHub repository (as mentioned in L173-174 of our main paper), our work goes further by enabling the handling of 6-DoF camera poses and gradient-based pose optimization. To achieve this, we fine-tuned Zero-123 with a novel 6-DoF camera parameterization and implemented customized CUDA kernels, as the official Gaussian Splatting codebase did not support this feature.
>
> We appreciate that the reviewer recognizes the novelty of our pose refinement strategy. However, we kindly ask the reviewer to also consider the broader contributions of our proposed system and the specific problem we aim to address. Specifically, we integrate gradient-based pose optimization with generative priors to tackle sparse-view pose refinement, a scenario prone to overfitting. Additionally, we introduced an outlier identification and correction system to manage significant initial pose errors so that our method can work in real-world scenarios where only estimated camera poses from off-the-shelf methods are available. To the best of our knowledge, no existing work effectively addresses this challenging yet practical setup.
>
> > The major weakness of the paper is the lack of fair comparisons in terms of the 3D reconstruction.
>
> Please refer to our general response, where we include an evaluation of SPARF regarding novel view synthesis. We apologize for not including these in our earlier version -- SPARF training is slow (the overall training takes more than 10 hours **per instance** on a single GPU such as V100) and due to limited resources, we only compared to SPARF's pose optimization stage (which takes 30% iterations). We also note that the suggested baseline DMV3D is not applicable in our setup -- it generates multiple views as output but does not allow multiple unposed views as input! Moreover, their implementation is not open-sourced.
>
> > I'm quite curious what the reconstruction quality the method can achieve without 3D generative prior but with the proposed refinement. Namely the combination of (1) and (4) in Table 4.
>
> We thank the reviewer for suggesting this experiment. We conducted the ablation study under the same settings as Table 4 and compared the results with our full system:
>
> | Full Method |  Rot@5°  | Rot@15°  |  CC@10%  |    F1   |   PSNR    |   LPIPS     |   PSNR*    |   LPIPS*    |
> | ----------- | :------: | :------: | :------: | :-------: | :--------: |:-------: | :--------: | --------- |
> | w/o SDS     | **66.2** |   86.8   |   77.8   | 63.57     | **16.05** | **0.2277** |   17.90   |   0.1990   |
> | w/ SDS      |   60.3   | **88.2** | **80.4** | **68.19** |   15.30   |   0.2304   | **18.18** | **0.1778** |
>
> In this setup, the quality of the 3D reconstructions is worse as there are more artifacts (significant roughness and holes) in the geometry. However, we found that this setup does achieve high pose accuracy (even being more precise on Rot@5° than our full system). Analyzing further, we found the no-SDS setup makes it easier to remove outliers (removing one input image more significantly reduces the reprojection errors). Quantitatively, **97.1%** of sequences in this setup involve at least one identified outlier, compared to **67.2%** in the full system.
>
> Perhaps more surprisingly, we found that this setup also yielded slightly better view synthesis results (PSNR and LPIPS), but this did not correspond to the qualitative results (see **Fig. 2** in our rebuttal PDF file). We hypothesize that this discrepancy is because we compute a single global alignment between our optimized camera poses and GT poses before evaluating NVS, and this alignment maybe slightly more precise for the baseline leading to improved NVS metrics even though the 3D/view-synthesis quality is worse. To mitigate the effects of alignment in NVS evaluation, we also report PSNR*, and LPIPS* -- where we locally optimize camera pose for each novel view for a few steps using photometric error while keeping our 3D representation fixed. We see that our full system does yield better predictions, highlighting that the use of SDS does indeed improve the novel-view generations.
>
> > How are the poses used for generative prior sampled in addition to the input views?
>
> We preprocess the initial camera poses provided by off-the-shelf methods to align the world origin with the approximate intersection of their optical axes. Next, we create a sphere centered at the origin, with a radius equal to the average distance of all cameras from the origin. When rendering novel views for SDS, we sample points on the sphere for camera translation using random azimuth and elevation angles. For rotation, we orient the sampled camera towards the origin.
>
> > How are the thresholds for pose outlier removal tuned?
>
> During the early stages of our work, we conducted initial experiments on a few samples from the CO3D dataset and found that a threshold of 0.05 for LPIPS generally worked well. This setting was maintained for all our in-the-wild evaluations on NAVI reported in the paper. We did not tune the threshold further for these results, though a fine-grained search might yield improved performance.

---

> > ### Comment · Reviewer_ePgG · 2024-08-09
> >
> > Thanks for the detailed reply. The authors' response addressed my major concerns about the evaluation and I decided to raise my rating to acceptance.

---

> > > ### Author Response · Authors · 2024-08-09
> > >
> > > We greatly appreciate the reviewer for raising the score! We will carefully revise our paper based on the reviewer’s valuable feedback. Thank you again for providing such insightful comments!

---

### Author Rebuttal · Authors · 2024-08-04

# General Response

We appreciate the reviewers' insightful comments and valuable feedback. We are glad that the reviewers appreciated the results, the practicality of the setup, and found the paper well written. In this response, we address some of the common points raised by the reviewers (additional NVS comparison and clarifications on inference), and address the more specific comments via separate responses to each review.

### Comparison with SPARF on Novel View Synthesis (NVS)

In addition to our submission comparing to SPARF for pose estimation, reviewers ePgG and XaBT recommended comparison to SPARF via NVS metrics. We report these below, using DUSt3R pose (N=6, 8) as initialization. Due to SPARF's long training time (about 10 hours per instance), we could only include 70 sequences (2 sequences per object) in these two experiments.

| N = 6     | Avg Rot Error | Rot@5°  | Rot@15° | CC@10%  | Improvement Rate | PSNR     | LPIPS     | *PSNR    | *LPIPS    |
|-----------|---------------|---------|---------|---------|--------------|----------|-----------|----------|-----------|
|   DUSt3R  |   7.90        |   48.5  |   93.3  |   80.2  |   /          |   /      |   /       |   /      |   /       |
|   SPARF   |   17.07       |   47.9  |   68.7  |   67.6  |   0.41       |   12.80  |   0.3201  |   15.30  |   0.2478  |
|   Ours    |   5.82        |   73.9  |   94.5  |   92.1  |   0.80       |   15.52  |   0.2179  |   16.23  |   0.1844  |

| N = 8     | Avg Rot Error | Rot@5°  | Rot@15° | CC@10%  | Improvement Rate | PSNR     | LPIPS     | *PSNR    | *LPIPS    |
|-----------|---------------|---------|---------|---------|--------------|----------|-----------|----------|-----------|
|   DUSt3R  |   8.71        |   51.6  |   92.9  |   79.5  |   /          |   /      |   /       |   /      |   /       |
|   SPARF   |   18.17       |   57.3  |   75.7  |   68.6  |   0.54       |   13.42  |   0.3059  |   15.44  |   0.2584  |
|   Ours    |   5.99        |   81.7  |   95.3  |   92.5  |   0.93       |   17.02  |   0.1874  |   17.48  |   0.1695  |

The results indicate that our method outperforms SPARF in pose accuracy and novel view quality. In fact, we found that SPARF can often make the poses worse compared to the (relatively accurate) DUSt3R initialization (measured via `Improvement Rate` that indicates the percentage of sequences with reduced pose error).  We found that this is because the correspondences leveraged by SPARF in its optimization are not robust and are susceptible to false matches -- please see **Fig. 3** of the rebuttal PDF for an example.

As an attempt to compare novel view quality despite the difference in pose accuracy, we report *PSNR and *LPIPS, which are measured *only on the sequences where SPARF improves pose accuracy* and find that even in these, our approach outperforms it. We also observed that while SPARF works well on novel views close to the input, floaters constantly appear with significant viewpoint changes. In contrast, our generative prior leads to a more consistent 3D representation.

## Inference Time and Details
### (1) Iterative Outlier Identification
Given N input images, our method first reconstructs 3D using the proposed MV-DG approach. The image with the largest reprojection error is considered as a candidate for being an outlier, and we perform another reconstruction after removing it (using remaining N-1 inputs). This candidate is considered a valid outlier if the average reprojection error in others views reduces significantly (we use 0.05 in LPIPS as a threshold). If it is an outlier, we remove it and repeat the process to identify outliers among the remaining N-1 images, stopping the process when no outliers remain. In general, if O outliers are detected, we need to perform O+2 reconstructions.

### (2) Outlier Correction via Render-and-compare
Any identified outlier in (1) requires pose correction. We do this by combining discrete search and continuous optimization. Specifically, we reuse the reconstructed 3D from (1) with only the inliers involved. For each outlier, we sample pose candidates evenly on a sphere around and targeted at the reconstructed 3D. Next, we optimize these pose candidates via gradient descent from the photometric error given our reconstructed 3D. To find out the most likely target pose, we select the optimized pose candidate whose rendering has the lowest discrepancy with the target input image, which is measured by both MSE and LPIPS. As MV-DG outputs a 3D mesh, rendering and propagating gradients to camera poses are fast. Each outlier takes roughly one minute to do pose correction.

If any outliers were corrected, another reconstruction is performed using the updated poses.

### (3) Inference time:
For N=8 images using a single RTX A5000, one reconstruction with MV-DG takes about 2 minutes to complete, and the 'render-and-compare' for each outlier takes around a minute. Our full pipeline (using RayDiffusion initialization) detected an average of 0.94 outliers per sequence, resulting in an inference time of around 9 minutes.

We will update the paper to include these details more clearly as well as release our implementation for reproducibility.

---

### Decision · Program_Chairs · 2024-09-25

**Decision:**

Accept (poster)

**Comment:**

The paper introduces a novel optimization-based method for sparse-view 3D reconstruction from unposed images. The method uses an off-the-shelf pose estimator for pose initialization, then optimizes both the pose and 3D reconstruction using rendering loss and generative priors. The method demonstrates satisfactory results on the evaluation data, and the ablation study highlights the effectiveness of each proposed technique. The initial ratings were three borderline accepts and one reject. The key concerns included a lack of novelty, unfair comparisons in terms of 3D reconstruction, increased inference time, and missing related work. In the rebuttal, additional 3D reconstruction comparisons were provided, and discussions on novelty, new related work, and inference time were included. The final ratings were four weak accepts. Given the consensus, the AC recommends acceptance.